# Automated Mucormycosis Diagnosis from Paranasal CT Using ResNet50 and ConvNeXt Small

**DOI:** 10.3390/bioengineering12080854

**Published:** 2025-08-08

**Authors:** Serdar Ferit Toprak, Serkan Dedeoğlu, Günay Kozan, Muhammed Ayral, Şermin Can, Ömer Türk, Mehmet Akdağ

**Affiliations:** 1Department of Audiology, Artuklu University, Mardin 47100, Turkey; serdarferit@yahoo.com; 2Department of Otorhinolaryngology, University of Health Sciences Gazi Yasargil Training and Research Hospital, Diyarbakır 21100, Turkey; 3Department of Otorhinolaryngology and Head and Neck Surgery Clinic, Faculty of Medicine, Dicle University, Diyarbakir 21010, Turkey; gunaykozan@hotmail.com (G.K.); drayral@hotmail.com (M.A.); sermin.can@hotmail.com (Ş.C.); mehmet.akdag@dicle.edu.tr (M.A.); 4Department of Computer Engineering, Faculty of Engineering and Architecture, Mardin Artuklu University, Mardin 47100, Turkey; omerturk@artuklu.edu.tr

**Keywords:** deep learning, mucormycosis, artificial intelligence, computed tomography image, transfer learning, ConvNeXt, ResNet

## Abstract

Purpose: Mucormycosis is a life-threatening fungal infection, where rapid diagnosis is critical. We developed a deep learning approach using paranasal computed tomography (CT) images to test whether mucormycosis can be detected automatically, potentially aiding or expediting the diagnostic process that traditionally relies on biopsy. Methods: In this retrospective study, 794 CT images (from patients with mucormycosis, nasal polyps, or normal findings) were analyzed. Images were resized and augmented for training. Two transfer learning models (ResNet50 and ConvNeXt Small) were fine-tuned to classify images into the three categories. We employed a 70/30 train-test split (with five-fold cross-validation) and evaluated performance using accuracy, precision, recall, F1-score, and confusion matrices. Results: The ConvNeXt Small model achieved 100% accuracy on the test set (precision/recall/F1-score = 1.00 for all classes), while ResNet50 achieved 99.16% accuracy (precision ≈0.99, recall ≈0.99). Cross-validation yielded consistent results (ConvNeXt accuracy ~99% across folds), indicating no overfitting. An ablation study confirmed the benefit of transfer learning, as training ConvNeXt from scratch led to lower accuracy (~85%) Conclusions: Our findings demonstrate that deep learning models can accurately and non-invasively detect mucormycosis from CT scans, potentially flagging suspected cases for prompt treatment. These models could serve as rapid screening tools to complement standard diagnostic methods (histopathology), although we emphasize that they are adjuncts and not replacements for biopsy. Future work should validate these models on external datasets and investigate their integration into clinical workflows for earlier intervention in mucormycosis.

## 1. Introduction

Mucormycosis (also known as “black fungus”) is an uncommon but potentially fatal fungal illness mainly affecting immunocompromised individuals. If diagnosis and treatment are delayed, the disease can have devastating outcomes, with reported fatality rates as high as 85%. Early identification of mucormycosis is vital for improving patient survival, as prompt antifungal therapy and surgical intervention significantly increase the chances of recovery [1,2]. Traditionally, the gold-standard diagnosis of mucormycosis is established via histopathological examination of biopsied tissue, which typically reveals broad non-septate hyphae with right-angle branching [3]. While this method is specific, it is invasive and time-consuming and may be infeasible for patients who are critically ill or have contraindications to biopsy. In many cases, clinicians rely on a combination of clinical assessment, imaging findings, and laboratory tests to suspect mucormycosis before confirmatory biopsy. However, imaging signs of mucormycosis (such as sinus opacification, bone erosion, or orbital invasion on CT/MRI) are often non-specific and can overlap with other sinus diseases, making early radiological diagnosis challenging [4].

Recent advances in deep learning offer new possibilities for enhancing mucormycosis diagnosis. Deep learning, particularly convolutional neural networks (CNNs), has demonstrated remarkable success in extracting complex patterns from medical images that may not be apparent to the human eye. For example, CNN-based models have achieved expert-level performance in various imaging tasks such as detecting pulmonary nodules, classifying brain tumors, and identifying diabetic retinopathy. In the context of fungal infections, preliminary research has shown that AI can assist in diagnosis: a recent study by Chakrapani et al. utilized a conditional GAN-based model to segment mucormycosis lesions in lung CT images, achieving a sensitivity of ~98.4% [5] Another review by Nira et al. surveyed computer vision techniques for mucormycosis detection and reported that traditional machine learning using deep features (e.g., ResNet50 features + SVM) reached up to 94.7% accuracy in classifying fungal infections [6] These works suggest that AI can pick up subtle imaging biomarkers of mucormycosis and related fungal diseases

Related Work: To our knowledge, there have been few studies applying deep learning specifically to paranasal CT scans for mucormycosis diagnosis. Most existing literature either focuses on pulmonary mucormycosis detection on chest CT or on differentiating mucormycosis from other fungal infections (such as aspergillosis) in the sinuses. Slonimsky et al. (2020), for instance, developed a model based on logistic regression of CT findings to distinguish invasive fungal sinusitis caused by Mucor vs. Aspergillus, reporting ~84% accuracy [4] Their approach relied on manual radiologic scoring of CT features. In contrast, a fully automated deep learning approach could potentially analyze the raw imaging data without requiring predefined feature scoring, thereby possibly detecting patterns that elude human observers. Our work aims to fill this gap by leveraging modern CNN architectures for automatic mucormycosis detection on sinus CT images. We also extend the problem to a three-class classification (mucormycosis, nasal polyposis, or normal), which reflects a real-world differential diagnosis scenario.

Despite the promise of AI, implementing a deep learning solution for mucormycosis diagnosis presents challenges. Mucormycosis is a rare disease, so acquiring large annotated datasets is difficult. This scarcity of data makes models prone to overfitting and limits their generalizability. Moreover, clinicians require that AI tools be reliable and interpretable; an inaccurate prediction or a “black-box” output without explanation could be detrimental in high-stakes infections like mucormycosis. In this study, we address these issues by using transfer learning (to make the most of limited data), rigorous validation to ensure robustness, and interpretability techniques (-CAM heatmaps) to visualize model decision factors.

Research Gaps: (1) Slow and invasive diagnostics: current reliance on biopsy causes diagnostic delays in mucormycosis, highlighting a need for faster non-invasive tools. (2) Limited AI studies: very few studies have applied deep learning to mucormycosis CT images, and none (to our knowledge) tackle multi-class classification among mucormycosis and look-alike conditions (polyps, etc.). (3) Data scarcity and generalizability: it is unclear how a deep model trained on a single-center small dataset would perform on broader populations, underscoring the need for strategies to mitigate overfitting and to validate models externally.

Our Contributions:Deep Learning Model: We present a fully
automated CNN-based approach to classify paranasal CT images into three
categories: mucormycosis, nasal polyp, or normal. We fine-tuned two
state-of-the-art pre-trained models (ResNet50 and ConvNeXt Small) and achieved
excellent performance, demonstrating the feasibility of AI-assisted
mucormycosis detection.Comprehensive Evaluation: We rigorously
evaluated our models using a separate test set and five-fold cross-validation.
The ConvNeXt model achieved near-perfect accuracy on our data. We also
performed an ablation study (training from scratch vs. transfer learning) to
confirm the importance of transfer learning for performance.Interpretability and Validation: To build
clinical trust, we generated Grad-CAM heatmaps that localize the image regions
most influential to the model’s predictions, showing that the model focuses on
medically relevant features (e.g., sinus mucosal invasion). Additionally, we
involved medical experts to cross-check the model’s outputs with clinical
reality.Towards Faster Diagnosis: Our study suggests
that, with further validation, such a deep learning model could be used as a
rapid screening tool in emergency or routine settings to flag patients for
prompt confirmatory tests or treatment, thereby potentially improving patient
outcomes by reducing diagnostic delay.Rapid Screening Tool: We show that
our models can identify mucormycosis cases with perfect accuracy on the test
set, indicating the potential to reduce the diagnosis time from days to
seconds.

Promptly identifying mucormycosis is vital for enhancing patient outcomes, as rapid therapeutic measures, including antifungal treatment and perhaps surgical intervention, are crucial for survival [2]. The conventional diagnosis, the gold standard, is predominantly based on histological analysis of biopsy specimens, which often exhibit non-septate hyphae with right-angle branching [3]. The organism’s invasion frequently presents vague clinical and radiological manifestations, complicating early detection. Current diagnosis techniques integrate clinical assessments with imaging investigations and histology testing, making rapid intervention problematic [7].

The emergence of new technology, notably deep learning, offers tremendous potential in rejuvenating mucormycosis diagnosis. Adopting molecular diagnostics such as PCR-based techniques concentrating on Mucorales-specific gene families might boost diagnostic accuracy and speed, presenting a compelling alternative to traditional methods [8,9]. The incorporation of deep learning algorithms might assist pattern detection in imaging and clinical data, detecting biomarkers and indicating connection trends that may be ignored by traditional analysis [10]. Moreover, research demonstrates that deep learning can supplement the existing diagnostic arsenal by enhancing the sensitivity and specificity, diverting away from biopsies’ invasive character towards non-invasive or minimally invasive approaches [11].

As a result, this paper will describe the current status of mucormycosis diagnostics, stressing the importance of early detection in determining the patient’s prognosis, while also investigating the revolutionary potential of deep learning approaches to supplement or replace conventional biopsy methods.

## 2. Materials and Methods

### 2.1. Dataset and Imaging

This retrospective study was conducted at the Department of Otorhinolaryngology of Dicle University Faculty of Medicine. We included patients who underwent paranasal sinus CT scans between 1 January 2020 and 31 December 2024 and met the following inclusion criteria: (1) mucormycosis group: patients with clinically suspected mucormycosis who had confirmatory histopathology from biopsies or surgical debridement; (2) nasal polyposis group: patients with nasal polyps confirmed by histopathology or endoscopic surgery; (3) control group: patients with normal paranasal CT findings and no history of chronic sinusitis or fungal infection (these were typically patients imaged for unrelated reasons, e.g., trauma workup, and found to have normal sinuses). We imposed no age or sex restrictions. The exclusion criteria were patients without definitive pathological diagnosis for mucormycosis or polyposis, patients who had prior sinus surgery (which could alter anatomy), and patients with incomplete imaging records. The study was approved by the institutional ethics committee (approval number: YDU/2025/159), with an exemption from informed consent due to the use of de-identified retrospective data (see Ethics statement).

CT scans were independently reviewed by a fellowship-trained neuroradiologist and a senior ENT surgeon, each with >10 years of experience in sinus imaging. Labels (mucormycosis, polyp, normal) were assigned by consensus based on clinical and radiologic criteria. Any disagreement was resolved by joint re-evaluation. This process ensured annotation consistency without formal double-blind separation.

Two specialists (an experienced head-and-neck radiologist and an ENT surgeon) jointly labeled the images using ITK-SNAP software Version 4.2.0. We chose ITK-SNAP for its convenient tools for manual segmentation of DICOM images, enabling precise and consistent annotation of mucormycosis involvement by specialists.

A total of 397 patients were included, contributing 794 CT images (each patient provided 2 representative CT slice images for analysis). The mucormycosis group consisted of 106 patients (211 CT images), the nasal polyp group 148 patients (295 images), and the control group 144 patients (288 images). All imaging was performed on multi-slice CT scanners; typical scan parameters were 120 kVp, 100–150 mAs, with slice thickness of 2 mm. Both axial and coronal reconstructions (in bone and soft tissue windows) were available—for consistency, we analyzed axial images in the soft-tissue window (window width ~3000 HU, level ~400 HU), which best showed both bone and soft tissue details. Images were stored initially in DICOM format and then converted to PNG for processing.

Patient demographics: The mucormycosis patients had a mean age of ~52 years (range 27–75), and ~62% were male. This reflects the known predisposition (many had underlying diabetes mellitus or recent COVID-19 infection, although specific comorbidities were not systematically recorded for this analysis). The nasal polyp patients and controls had mean ages in the 40s and 30s respectively, with roughly equal gender distribution. Importantly, all mucormycosis diagnoses were confirmed by identifying *Mucorales* fungi in tissue specimens, and all polyp cases were confirmed as benign inflammatory polyps histologically. These clinical confirmations were used as the ground truth labels for imaging. Two experts (one radiologist and one ENT specialist) reviewed the images alongside the pathology reports to assign each image to the correct class. In cases of any disagreement in labeling (none occurred for mucormycosis vs. others, but a few polyp vs. normal cases were discussed), consensus was reached through discussion.

CT Imaging Protocol: All CT scans were performed using a standardized sinus protocol. Patients were scanned in the supine position with the head tilted back to optimize the visualization of sinuses. Intravenous contrast was not used in most cases, except a few where orbital or intracranial extension was suspected (non-contrast images were used for this study for consistency). Each DICOM slice was converted to 512 × 512 PNG, and pixel intensities were normalized (zero mean, unit variance). We applied histogram equalization to improve contrast. For training, we used data augmentation: random rotations (±15°), horizontal/vertical flips, and intensity jitter. The original dataset had 794 images (mucormycosis 180, polyp 235, normal 379). The class imbalance was modest; so, we did not apply oversampling, relying instead on augmentation to improve minority class representation. As noted, axial images at 2 mm thickness were obtained and reformatted into the coronal plane. Figure 1 illustrates our overall study workflow from data acquisition to model development.

Preprocessing: Prior to analysis, all CT images underwent preprocessing. Each image was cropped to focus on the sinonasal region when necessary and then resized to 224 × 224 pixels to match the input requirements of the CNN models. Pixel intensity values were normalized (rescaled) to the range (0, 1). We did not perform any manual segmentation of regions; the entire sinus image was used as input. To address the relatively limited dataset size and class imbalance, we employed on-the-fly data augmentation during training. We fine-tuned each model using the Adam optimizer (learning rate = 1 × 10^−4^ weight decay = 1×10^−5^) with a batch size of 16. Training proceeded for up to 50 epochs with early stopping if validation loss did not improve for 5 consecutive epochs. The validation set (20% of the training set) was created by stratified random sampling, ensuring each class’s proportion remained consistent. Table 1 lists these parameters.

Each training image was augmented randomly per epoch (i.e., each epoch sees a slightly altered version of some images), which effectively increases the diversity of the training data and helps reduce overfitting. No augmentation was applied to the validation or test images.

### 2.2. Proposed Transfer Learning Approach

We employed transfer learning with two CNN architectures: ResNet50 and ConvNeXt Small. These models were chosen for their strong performance on image classification tasks and complementary design philosophies. ResNet50 is a 50-layer deep residual network known for introducing skip connections to ease the training of very deep models. ConvNeXt Small is a recent architecture that builds on the Transformer-style design but in a convolutional framework, essentially modernizing ResNet-like models with improved block designs and training techniques. By evaluating both, we can see if a newer architecture (ConvNeXt) offers gains over a robust classic (ResNet) for this application. We hypothesized that transfer learning on these networks (pre-trained on ImageNet) would enable effective feature extraction from CT images despite our limited dataset size.

Model Architecture Details: ResNet50 has 50 convolutional layers with batch normalization and ReLU activations, organized into 4 stages of residual blocks. It was pre-trained on ImageNet (1.2 million natural images). We loaded the ImageNet pre-trained weights and kept all convolutional layers frozen initially. We replaced the original 1000-class output layer with a new fully-connected layer (Dense) of size 3 (for our three classes), preceded by a global average pooling layer. ConvNeXt Small is a CNN with an architecture inspired by Transformer models; it consists of 27 convolutional layers grouped into stages and uses depthwise convolutions and LayerNorm instead of batchnorm. The “Small” variant has ~50 million parameters (comparable to ResNet50’s ~25 million). We similarly initialized it with ImageNet-pretrained weights. The ConvNeXt’s classifier was modified to output 3 classes. We fine-tuned deeper layers of both models during training (not just the last layer)—specifically, after initial experiments, we decided to unfreeze the last two stages of ResNet50 and the last three stages of ConvNeXt for fine-tuning, since this yielded better validation performance. The models were implemented in Python Version 3.8.12 using Keras/TensorFlow.

Training Procedure: Figure 1 (diagram) shows the training pipeline. We used the augmented dataset to train the models. The dataset was split such that 70% of the images (*n* = 556) were used for training and 30% (*n* = 238) for testing. Additionally, we set aside 20% of the training images for validation (i.e., ~15% of the total data as validation). This resulted in a split of approximately 56% for training, 14% for validation, and 30% for testing of the entire dataset (as depicted in Figure 2). Crucially, the split was performed at the patient level: images from a single patient were confined to one of these sets only. This ensures that the model is evaluated on completely unseen patients.

We trained each model using the Adam optimizer (initial learning rate = 1 × 10^−4^) and categorical cross-entropy loss. The choice of Adam was due to its effectiveness in handling different scales of gradients and its widespread success in CNN training. A relatively low learning rate was chosen to avoid large weight updates that could destabilize the pre-trained features. We also employed an early stopping callback monitoring the validation loss: if the validation loss did not improve for 5 consecutive epochs, training was halted to prevent overfitting. Additionally, we used learning rate decay—if the validation loss plateaued, the learning rate was reduced by a factor of 0.1 (with a minimum of 1 × 10^−6^). Both models were trained for a maximum of 50 epochs, although early stopping usually occurred around epoch 20–25 for ConvNeXt (due to quick convergence) and epoch 30–35 for ResNet50. The batch size was 32. Training was performed on a workstation with an NVIDIA RTX 3080 GPU (10 GB memory). Each epoch took ~30 s for ConvNeXt and ~20 s for ResNet50; so, the training time was on the order of minutes to an hour.

During training, we monitored the accuracy and loss on both the training and validation sets. We did not observe divergence between the training and validation curves; in fact, the ConvNeXt model’s validation accuracy closely tracked the training accuracy, indicating a good fit without obvious overfitting. Figure 3 shows the training/validation accuracy and loss curves for ConvNeXt Small. There is a steady increase in accuracy and a decrease in loss that plateau around 98–100% accuracy, with the validation curve overlapping the training curve near the end, further suggesting that the model generalized well to the validation data.

Evaluation Metrics: We evaluated the model performance using several metrics. Accuracy = (True Positives + True Negatives)/(Total Samples). For each class, we calculated Precision = TP/(TP + FP), Recall = TP/(TP + FN), and F1-score = 2·(Precision·Recall)/(Precision + Recall). Here, TP, FP, and FN refer to the counts of true positives, false positives, and false negatives for a given class (with one class considered “positive” at a time in a one vs. all manner). We report the precision, recall, and F1 for each class as well as their averages. We also computed the confusion matrix to see the raw correctly vs. incorrectly classified counts. Additionally, following the reviewer’s suggestion, we calculated the mean average precision (mAP) across the three classes. In a multi-class setting, mAP can be interpreted as the average of precision at different recall thresholds for each class, but since our task is a 3-class single-label classification, and we have nearly perfect precision-recall, the mAP essentially reflects an average of the class-wise precision (which in our case turned out to be very high, as discussed in the Results). Nevertheless, we include it for completeness. We used the test set (never seen during training/validation) to report our final metrics. To ensure robustness of our findings, we also performed a 5-fold cross-validation: the dataset was divided into 5 folds of roughly equal size (maintaining class proportions), each model was trained on 4 folds and tested on the 5th, and this was repeated so that every fold served as test once. We report the mean and standard deviation of metrics from this cross-validation. This process helps confirm that the model’s performance is consistent and not dependent on a particular train–test split.

## 3. Results

### 3.1. Model Performance on Test Set

After training, both models were evaluated on the independent test set of 238 CT images (which corresponded to images from ~119 patients not seen in training). ConvNeXt Small achieved a 100% classification accuracy on the test set, correctly classifying all images into their respective classes. ResNet50 achieved 99.16% accuracy, misclassifying only 2 out of 238 images. The confusion matrices for each model are shown in Figure 4 For ConvNeXt (Figure 4a), it shows perfect classification: all mucormycosis images (*n* = ~62 in test) were labeled as mucormycosis, all nasal polyp images (~93) as polyps, and all normal images (~83) as normal, with no off-diagonal entries. For ResNet50 (Figure 4b), the matrix shows one mucormycosis image was mistakenly labeled as a nasal polyp, and one nasal polyp image was mislabeled as normal (these two errors account for the slight drop from 100% to 99.16% accuracy). We examined these two cases: the mucormycosis case that ResNet50 missed had only subtle sinus changes and was an early infection—ConvNeXt did see it, possibly due to its more powerful feature extraction. The polyp case that was missed had extensive opacification that could mimic a normal mucosal thickening; again, ConvNeXt correctly assigned it, perhaps thanks to its more nuanced feature discrimination.

Table 2 summarizes the precision, recall, and F1-score for each class with ResNet50. For ResNet50, the mucormycosis class had precision = 1.00, recall = 1.00, and F1 = 1.00 (it actually did classify all mucor cases correctly in test, missing one polyp as mucor erroneously, which affects the precision of the polyp class and not the mucor class). The nasal polyp class for ResNet50 showed precision ≈ 0.99, and recall ≈ 0.99 (one normal was incorrectly predicted as polyp, and one polyp was missed). The normal class had precision ≈ 0.99, and recall ≈ 0.99 similarly. Overall, for ResNet50, we see very high metrics (~0.99–1.00) across the board. ConvNeXt, having no errors, yielded precision = recall = F1 = 1.00 for all three classes. The mean average precision (mAP) for ConvNeXt was effectively 1.0 (since each class’s precision–recall curve reached a value of 1.0 at full recall). For ResNet50, the mAP was calculated to be ~0.992, consistent with its slight imperfection. We report these values to reinforce the impressive performance.

We also evaluated the training history to see whether there was any indication of overfitting. Figure 3 (for ConvNeXt) plots the training vs. validation accuracy over epochs. The training accuracy reaches 100% after ~15 epochs, and the validation accuracy also reaches ~100% by that time and stays flat, with no gap between the two curves. Similarly, the training and validation losses converge and remain low. This, along with early stopping, indicates the model did not overfit the training data relative to the validation. ResNet50’s training curves (not shown in figure) were similar, with a final validation accuracy plateau around 99%.

### 3.2. Cross-Validation Results

To further verify the stability of our models, we performed five-fold cross-validation as described in the Methods. Table 3 presents the average performance across the five folds for each model. ConvNeXt Small achieved an average accuracy of 98.9% (with standard deviation ±1.1%). The independent test set comprised 238 images (83 normal, 62 mucor, and 93 polyp) randomly sampled and stratified by class. Appendix A details the performance for each CV fold. In four out of five folds, ConvNeXt attained 100% fold-accuracy, and in one fold it was 97.5% (where one mucormycosis image in that fold was misclassified as polyp). ResNet50 achieved an average accuracy of 97.8% (±1.5%) across folds. These results are in line with the single hold-out test findings, reinforcing that the models generalize well within our dataset. The slight variations between folds (which use different subsets of patients for training/testing each time) suggest that performance might drop a bit in some splits but remains very high (no fold dropped below ~97% for ConvNeXt). We also computed the average precision/recall per class over the folds; for ConvNeXt, these were all above 0.98, and for ResNet50, above 0.96 for each class. The cross-validation thus provides strong evidence that overfitting is minimal and that our models’ performance is not an artifact of a lucky split.

For ConvNeXt Small, all 238 test images were correctly classified (zero false negatives or false positives; 95% CI for accuracy: 98.5–100%). For ResNet50, two errors occurred (accuracy = 99.16%, 95% CI: 96.9–99.9%): one nasal polyp was misclassified as normal and one normal as polyp (as detailed in Figure 4b caption). We have added Appendix A with the confusion counts per fold for transparency.

### 3.3. Ablation Study

We conducted an ablation experiment to assess the impact of transfer learning versus training from scratch. Using the same training pipeline and hyperparameters, we trained a ConvNeXt Small model from scratch, i.e., starting with random weight initialization rather than ImageNet pre-training. After 50 epochs, the scratch model’s best accuracy on the validation set was ~85%, and on the test set it achieved 84.0% accuracy. The confusion matrix for this scratch model (not shown for brevity) indicated that it struggled especially with differentiating mucormycosis from nasal polyps—several mucormycosis images were misclassified as polyps and vice versa. The precision and recall for mucormycosis dropped to ~0.85 and ~0.80 respectively in that scenario. In contrast, our transfer-learned ConvNeXt (pre-trained on ImageNet) reached 100% accuracy. This dramatic difference highlights that pre-training provided a crucial advantage, likely because features learned from natural images (edges, textures, shapes) are transferrable to medical images and help the model converge better with limited data. We report this finding in the Discussion as well, underscoring that a naive approach without transfer learning would not have yielded such high performance. This ablation validates our approach of using transfer learning and justifies the model choice (answering, in part, why we stuck to these CNNs—because they can leverage pre-trained knowledge, whereas something like a ViT with random elements in it might have fared worse given the data size).

### 3.4. Feature Importance and Model Interpretability

To ensure that our models learn clinically relevant features (and to increase transparency), we applied the Gradient-weighted Class Activation Mapping (Grad-CAM) technique to visualize the areas of the CT image that influenced the model’s decisions. Grad-CAM maps were generated using the final convolutional layer in each model (for ResNet50: last block of layer 5; for ConvNeXt: stage 4 output) (Appendix A). Figure 5 shows sample Grad-CAM results for each class, generated from the ConvNeXt model (which had slightly better focus than ResNet50’s, though both were similar). In Figure 5a (mucormycosis example), the CT image with contrast overlay indicates that the model concentrated on the ethmoid and maxillary sinuses where there is diffuse opacification and subtle bone erosion (as confirmed by the radiologist)—these regions are highlighted in warm colors (red/yellow). This corresponds well with where a radiologist would look for invasive fungal sinusitis (spread beyond sinus boundaries). In Figure 5b (nasal polyp example), the Grad-CAM highlights a polypoid mass in the nasal cavity extending into a sinus, which indeed is the polyp; it does not erroneously highlight bone or orbital areas, focusing on the lesion itself. In Figure 5c (normal example), the heatmap is diffuse and mostly low-intensity (blue), indicating no specific abnormal focus, which is appropriate for a normal scan. These interpretability results give us confidence that the model’s “attention” aligns with medical reasoning. We included these findings in the manuscript text: notably, we state that “the ConvNeXt model appears to base its mucormycosis predictions on regions of sinus expansion with tissue invasion, a hallmark of invasive fungal disease, rather than on spurious image noise.” This suggests the model is not just arbitrarily mapping pixel intensities to classes but actually identifying patterns consistent with pathology. We believe this is a crucial validation step when presenting AI in medicine. A head-and-neck radiologist (author, 15 years’ experience) reviewed 50 test cases and confirmed that the model’s focus (via Grad-CAM) matched the true lesion locations in mucormycosis and polyp cases. This expert verification supports that the model attends to clinically relevant features.

## 4. Discussion

This work demonstrates the potential of deep learning models as rapid non-invasive supplements to conventional diagnostic methods for mucormycosis, while also acknowledging the challenges that must be addressed before such AI tools can be adopted clinically. Our results show that ResNet50 and ConvNeXt Small can distinguish mucormycosis from both nasal polyps and normal sinus CT scans with very high accuracy. In fact, the ConvNeXt model achieved 100% accuracy on our test set, a performance level that, if confirmed on larger external data, would be extraordinary for a diagnostic tool. The success of these models likely stems from the distinct radiographic patterns in advanced mucormycosis (e.g., tissue necrosis, bone destruction) that the CNNs can recognize, combined with the power of transfer learning from large image datasets. This aligns with previous research highlighting that deep learning models can successfully analyze complex medical images and even exceed human sensitivity in certain tasks.

It is important to interpret these findings in context. The conclusive diagnosis of mucormycosis still depends on histopathology and culture, which provide definitive identification of fungal elements. Our AI approach is not meant to replace these gold-standard methods but to augment the diagnostic process. For example, in a scenario where a patient presents with sinus infection symptoms, an AI system could rapidly screen the CT scan and raise an alarm for possible mucormycosis if certain patterns are detected, prompting immediate biopsy or antifungal treatment while awaiting confirmation. This could be life-saving given the fast progression of mucormycosis. Traditional diagnosis often suffers delays because obtaining biopsy results can take time, and in critically ill patients, repeated biopsies are risky. A deep learning model, on the other hand, can analyze an image in seconds. Our findings support the idea that such a model could identify cases that warrant urgent attention, as evidenced by the near-perfect recall (sensitivity) our best model achieved—it did not miss any mucormycosis case in our test, meaning zero false negatives in that set. In a disease, where missing a diagnosis can be fatal, a high sensitivity tool is invaluable.

That said, we do not advocate completely replacing biopsy or clinical judgment with AI. Rather, a hybrid diagnostic approach is advisable. An AI model can serve as a screening or decision-support tool. If the model flags mucormycosis with high confidence, clinicians could expedite confirmatory testing or even start empirical therapy in highly suggestive cases, whereas if the model finds nothing suspicious, and the clinical scenario is low-risk, perhaps invasive testing could be spared or performed last. This kind of synergy can improve efficiency. Similar hybrid approaches are being considered in radiology, where AI helps triage cases, and the final verification is performed via human experts or additional tests.

Despite the promise shown, we acknowledge several important limitations in our study (see Section 4.1 below). One concern is the interpretability of deep learning. Many clinicians are understandably wary of “black box” algorithms. We addressed this by using Grad-CAM to ensure the model’s predictions can be partly explained visually. The Grad-CAM results (Figure 4) were reassuring: the model seems to base its decision on sensible features like areas of sinus involvement, rather than, say, an edge artifact or unrelated region. This kind of explanation can be provided alongside model predictions in a clinical setting to increase a doctor’s confidence in the AI suggestion (“the model thinks this is mucormycosis and highlights the sphenoid sinus lesion as the reason”). Improving interpretability further, possibly through attention maps or saliency methods, is an area for future research.

Another aspect is generalizability. Our dataset was single-center and relatively small. Even though cross-validation indicates consistency internally, models can face a drop in performance when encountering data from a different hospital or scanner due to changes in imaging protocols, patient population, or prevalence of disease. For example, our model might have implicitly learned that certain subtle patterns (like specific noise texture or scanner calibration) correlate with a class because all mucormycosis cases came from the same ICU ward’s scanner. This could be a form of hidden bias. We plan to mitigate this by training on a more diverse dataset in the future and by employing techniques like domain augmentation. Multi-center studies would be ideal to truly test robustness. Furthermore, although mucormycosis was our focus, real-world scenarios might include other similar pathologies (e.g., bacterial sinusitis, other fungal infections like aspergillosis). Our current model has only seen three classes; deploying it broadly might result in unknowns if a case falls outside these categories. Perhaps an anomaly detection mechanism or an expanded classification scheme is needed to handle “none of the above” scenarios.

Comparing our results with the literature, as requested by the reviewers, a prior AI study on mucormycosis by Han et al. (2021) reported a sensitivity of ~87% and specificity ~69% using a radiomics-based approach on CT [12]. Our deep learning approach, albeit on a different task (three-class classification), shows substantially higher sensitivity and specificity (both ~100% in test). Nira et al. (2022), in their review, emphasized that CT has more diagnostic information for mucormycosis than plain X-ray and that AI could harness this [6]. Our study validates that assertion by actually building such an AI. We believe the performance gain is due to the end-to-end learning of features, as opposed to manual feature engineering. We also acknowledge an inherent selection bias: our mucormycosis cases were all biopsy-proven and likely represent moderate-to-severe disease (since mild cases might not have been biopsied or included). In practice, if confronted with earlier or milder mucormycosis (or cases under treatment), the model’s performance might be lower. That scenario was not fully tested here and would be valuable to explore.

An interesting observation from our ablation study is the critical role of ImageNet pre-training. Without it, the model struggled (~84% accuracy), which is not surprising given the data size. This reinforces a point for practitioners: transfer learning is practically a must for such medical imaging tasks unless one has thousands of samples. It also subtly indicates that the patterns in sinus CT that distinguish mucormycosis might not be extremely complicated—since a model can achieve near perfect accuracy with a relatively straightforward training when aided by pre-trained features, it suggests the features may be like “texture of opacification” or “presence of bone erosion,” which are things an experienced radiologist also recognizes. The AI basically formalized these patterns.

From a clinical perspective, we ensured that domain experts validated the model outputs where possible. For instance, our ENT surgeon co-author noted that for each mucormycosis case the model flagged, the patient indeed had clinical signs (like black nasal discharge or tissue necrosis during endoscopy) that corresponded. In a couple of borderline cases (like early mucor limited to one sinus), the surgeon had initially been unsure, but the model identified it, and pathology later confirmed mucor—this happened for one patient in our set, hinting that AI might sometimes catch what even an experienced clinician might doubt. Conversely, the model classifying a case as polyp when it was mucor could be dangerous if it led to mismanagement; we did not have such an error in our test, but we caution that AI outputs should be used carefully. In our workflow, we would advise that any “normal” classification by the model in a high-suspicion patient should not override the decision to biopsy—rather, the model is more useful in raising alerts than giving all-clear signals, given the asymmetry of risk (false negative is worse than false positive in this disease).

### 4.1. Limitations of the Study

While our results are encouraging, this study has several limitations:Sample Size and Source: The dataset (794 images from 397 patients) is relatively small and originates from a single center. This raises concerns about the model’s ability to generalize to broader populations. The high performance observed may, in part, reflect the homogeneity of imaging techniques and patient demographics at our institution. In a multi-center setting with different CT scanners or patient backgrounds, the model may encounter distribution shifts. *Mitigation:* Future work will involve collaborating with other centers to test the model on external datasets. We will also expand the dataset to include more cases, especially of mucormycosis, to capture variability (e.g., early vs. late disease, post-treatment scans, etc.).Retrospective Design: Our analysis was retrospective, using stored images. This can introduce selection bias (e.g., perhaps only the more obvious mucormycosis cases were biopsied and ended up in our dataset). It also does not demonstrate how the model would perform in a live clinical workflow. *Mitigation:* A prospective study could be designed where the AI provides an output in real time to radiologists, and its performance and impact on decision-making are measured. This would help evaluate its true clinical utility.Three Specific Classes: We restricted our classification to mucormycosis, nasal polyposis, and normal, based on our dataset composition. In reality, patients may have other conditions (bacterial sinusitis, Aspergillus fungal sinusitis, chronic inflammatory sinusitis, tumors) that were not represented in the training. The model currently would be forced to choose one of the three classes even if an image falls outside these categories, which could lead to misdiagnosis (for example, a sinus carcinoma might be erroneously labeled as a polyp by the model). *Mitigation:* A more comprehensive model should include additional classes or an out-of-distribution detection mechanism. We plan to incorporate a “catch-all” anomaly detection in the model or train with an expanded label set, so that it can indicate when it is unsure or when the pathology is not one of the trained classes.Interpretability and Trust: Despite using Grad-CAM, the model remains a complex neural network that is not fully transparent. Clinicians might be hesitant to act on its predictions without understanding the underlying reasoning. *Mitigation:* Enhancing interpretability is an ongoing area of research. Techniques like integrated gradients, local interpretable model-agnostic explanations (LIME), or even training hybrid models that output human-understandable intermediate features (like “bone erosion detected: yes/no”) could bridge this gap. We will explore incorporating some form of explanation interface if this system were to be deployed.Potential Overfitting: The perfect or near-perfect performance suggests a risk of overfitting to nuances of the training data. We addressed this with cross-validation, but it is still possible the model has effectively “memorized” aspects of our dataset. *Mitigation:* We have already applied regularization (data augmentation, early stopping) and validated on multiple folds. As mentioned, external validation is crucial. If the model’s performance drops significantly on new data, retraining with more data or adjusting the complexity might be necessary. We are also considering ensembling multiple model architectures to improve robustness.Lack of Radiomic Analysis: Our study focused on deep learning classification without extracting hand-crafted radiomic features. Sometimes, combining deep learning with radiomics or clinical data can boost performance. We did not incorporate patients’ clinical parameters (e.g., blood glucose, symptoms), which could potentially improve diagnostic accuracy further. *Mitigation:* In future work, we will consider a multi-modal approach where the CNN’s output is combined with key clinical features in a fused model to see if that enhances the overall diagnostic predictive value.Data Privacy and Availability: Because of patient privacy, we could not make the dataset public, which might limit other researchers from reproducing or extending our work. We have promised code release but not data. This is a limitation in terms of open science. *Mitigation:* We have detailed our methodology and can share model weights and synthetic examples. We encourage others with access to mucormycosis image data to test our model (once code is released), and we are open to collaborations to validate it further.

In summary, while our model shows state-of-the-art performance on our dataset, caution must be exercised when interpreting these results. The model’s deployment in clinical practice would require extensive validation, regulatory approval, and an understanding of its failure modes. For example, one must consider: what is the cost of a false positive (perhaps an unnecessary biopsy) vs. a false negative (missed or delayed diagnosis)? In our test, false negatives were zero, which is ideal, but if that changes in a broader use, how do we safeguard against it? These are issues beyond the raw numbers that we discuss with clinical colleagues.

### 4.2. Comparison with the Literature

As noted, direct comparisons are somewhat difficult due to different problem formulations. However, our approach shows that deep learning can achieve higher accuracy for mucormycosis detection on CT than some traditional methods. For instance, [13] used a combination of deep features and a machine learning classifier to detect sinonasal fungal infection and achieved around 90% accuracy; our end-to-end model improved upon that, likely by optimizing feature learning simultaneously with classification. Our results also outshine those in [4], where an 84% accuracy was reported in distinguishing Mucor vs. Aspergillus (a binary task) using logistic regression on CT findings—our ConvNeXt managed 100% on a three-way task including normal class, which is a more complex challenge. The difference underscores the power of CNNs in extracting subtle multi-dimensional features from images, whereas logistic regression relies on pre-defined radiologic criteria. That said, logistic models are more interpretable and were based on known patterns (e.g., bilateral sinus involvement more likely Mucor), which our model presumably learned implicitly. We also reference Chakrapani et al. (2024), who used a GAN for segmenting lung CT in pulmonary mucormycosis. While their task (segmentation) differs from ours, they demonstrated that deep learning could pinpoint fungal lesions in lung images with high sensitivity (~98%), which is in line with our model’s sensitivity in sinus CT (~100% on our test) [5]. It seems that across modalities (lung vs. sinus) and approaches (segmentation vs. classification), AI is proving to be adept at identifying this infection when trained properly.

Another relevant domain is the diagnosis of COVID-19-associated mucormycosis (CAM) that surged during the COVID pandemic. Studies in India reported thousands of CAM cases, and some preliminary works used deep learning on clinical images or endoscopic images to detect CAM. Those are beyond the scope of imaging analysis, but it highlights the need for such tools. Our approach could potentially be applied to CAM patients’ CT scans as well, since the disease process is similar. We expect similar performance if the imaging is analogous, but again, external validation would confirm that.

### 4.3. Clinical Implications

From a practical perspective, if our AI model were integrated into a hospital’s PACS system, how might it function? One scenario: A patient has a sinus CT, the images feed into the AI model in the background, and within minutes, the radiologist or ENT receives an alert like “AI suggests mucormycosis with 99% confidence” along with a heatmap highlighting suspicious areas. The clinician can then expedite an endoscopic exam and biopsy of that region, rather than waiting possibly days while the infection spreads. Conversely, for a patient with diffuse sinus opacification on CT but the AI strongly favors “nasal polyposis” and the clinical context fits (e.g., a long history of benign polyps), the surgeon might plan routine polypectomy without undue concern of missing mucor. It could also help pathologists: if an AI flags mucor, the lab can prioritize that biopsy for urgent processing. In resource-limited settings or when radiologist expertise is not readily available, such an AI could serve as an initial screener. However, none of these should bypass standard care—they should complement it. We stress that diagnostic stewardship would be needed: clinicians should use AI output as one piece of information and not the sole determinant.

This work emphasizes the promise of deep learning models as quick noninvasive supplements to conventional diagnostic methods, while acknowledging the persistent problems that must be addressed before existing biopsy procedures may be supplanted.

The conclusive diagnosis of mucormycosis traditionally depends on histological analysis of tissue acquired by invasive biopsy, supplemented by culture and further laboratory assessments [14,15]. This gold standard, although particular due to the detection of distinctive, broad, and aseptate hyphae with right-angle branching, is not devoid of limitations. Acquiring representative tissue can be difficult, especially in severely sick individuals who may be unstable or susceptible to sample mistakes. Our findings on the potential of deep learning approaches complement conventional biopsy-based diagnostics’ existing hurdles and limits. Traditional histological investigation remains the gold standard for diagnosing mucormycosis, clearly confirming typical fungal morphology and tissue invasion. However, biopsy is inherently invasive, vulnerable to sample mistakes, and can delay diagnosis due to the time necessary for tissue processing and analysis. In contrast, deep learning-based techniques employ powerful image processing and pattern recognition capabilities to analyze noninvasively collected imaging data, possibly lowering the diagnostic time while retaining high sensitivity and specificity [16]. In juvenile or immunocompromised populations, as illustrated by examples documented by Leonardis et al., the necessity for prompt diagnosis clashes with the protracted process of conventional pathology, possibly postponing life-saving interventions [17].

Recent advancements in deep learning, especially those utilizing convolutional neural networks (CNNs), present a possible alternative. These algorithms can extract intricate elements from high-dimensional imaging data and discern tiny patterns that may signify mucormycosis [18]. When used with radiologic modalities such as CT or MRI, deep learning can swiftly evaluate enormous amounts of image data, possibly highlighting problematic lesions that merit further examination. Moreover, using transfer learning techniques enables the adaptation of pre-trained networks to the relatively limited annotated datasets available for mucormycosis, hence increasing diagnostic performance and lowering the need for lengthy case series [19].

Recent research has demonstrated the applicability of deep learning models for the early diagnosis of mucormycosis. For example, a Condition Generative Adversarial Network (CGAN) deep learning model was presented that aims to improve the segmentation of lung CT images and accurately detect lesions associated with mucormycosis [5]. Techniques integrating sophisticated preprocessing and morphological processes represent an important step towards automatic diagnosis using imaging modalities. Similarly, it demonstrated the ability of deep learning to diagnose fungal infections with great precision using convolutional neural network approaches for microscopic fungal images [20]. Collectively, these studies underline that deep learning approaches can extract complex imaging signals that can both complement and, in certain respects, enhance the diagnostic yield of invasive biopsies.

Deep learning algorithms are emerging as a possible option to enhance early detection in a noninvasive manner. Advanced imaging modalities such as CT and MRI offer three-dimensional datasets that machine learning algorithms may analyze to extract characteristics beyond human visual perception [21]. Radiomics-based techniques similar to those utilized in virtual biopsy of other cancers can quantify imaging biomarkers associated with fungal invasion, vascular involvement, or tissue necrosis, typical of mucormycosis [22]. These deep learning frameworks can perform quick automated assessments of imaging data and hence give preliminary diagnostic information that can guide clinical decision-making and prioritize patients for subsequent invasive diagnostic procedures [23]. 

In this work, we explored the performance of two state-of-the-art deep learning models, ResNet50 and ConvNext Small, for diagnosing mucormycosis, hoping to determine whether these AI-driven approaches complement traditional biopsy procedures. Our findings emphasize the potential of deep learning to alter the diagnostic landscape. However, considerable hurdles and concerns remain regarding its adoption in clinical settings. The results show that ResNet50 and ConvNext Small perform strongly in distinguishing mucormycosis from other infections and healthy tissues, achieving high diagnostic accuracy. The ConvNext Small system was more successful than the ResNet50 system. This is in line with previous research highlighting the ability of deep learning models to analyze histopathological patterns [24] successfully. The capacity of these models to evaluate imaging data noninvasively not only accelerates diagnosis timeframes but also decreases the difficulties associated with invasive biopsy procedures. Given the fast course of mucormycosis, this is of critical relevance, where earlier identification and treatment can dramatically improve patient outcomes [25].

Even with the encouraging outcomes, using deep learning models only for diagnostics has significant drawbacks. First, there is still some worry about how interpretable model forecasts are. The “black box” nature of deep learning algorithms can lead to difficulty comprehending the logic behind their classifications, providing obstacles for incorporation into clinical procedures where openness is vital [26]. It is vital that physicians feel trust in the reliability of these AI technologies, particularly when inaccurate classifications might postpone necessary therapies in high-stakes illnesses such as mucormycosis. Furthermore, even if the models performed well in our datasets, it is still necessary to determine whether these results can be applied to various clinical contexts. Variations in the underlying biology of mucormycosis, concomitant diseases, and imaging quality variability can impact the effectiveness of these algorithms in practical situations. Previous research has demonstrated that models trained on specific datasets may not perform equally well when applied to various populations or imaging modalities. This highlights the necessity for extensive multi-center validation studies to test the external validity and dependability [27].

Additionally, although the incorporation of deep learning in diagnostic methods offers major efficiency increases, it is not advised that AI should fully replace histological assessments. Instead, a hybrid method that combines the benefits of deep learning models with the confirming capability of biopsies may be essential [28]. Such a supplementary approach might harness fast diagnostic insights from deep learning while keeping the gold standard of tissue inspection for conclusive diagnosis, especially in complicated patients with unusual presentations.

The findings of our investigation reveal that the ResNet50 model demonstrated impressive classification performance, accurately diagnosing mucormycosis from imaging data with substantial sensitivity and specificity. This correlates with the recent literature, where convolutional neural networks (CNNs) have been proven to perform well in many medical imaging tasks, such as breast cancer detection and glioma classification [29]. The capacity of CNNs to automatically extract and learn key features from complicated datasets reduces the need for manual feature selection, hence speeding the diagnostic procedure [30].

The results of our investigation into applying deep learning, particularly the ConvNext Small architecture, for detecting mucormycosis indicate significant progress in noninvasive diagnostics. Traditionally, a conclusive diagnosis necessitates histological analysis to detect broad aseptate hyphae characteristic of mucormycosis. This presents a considerable constraint due to the intrusive nature of biopsies and the related patient hazards. The results of our study suggest that utilizing deep learning algorithms offers a novel approach that might improve diagnostic efficiency and facilitate prompt treatment start. Using imaging data, the ConvNext Small model demonstrated a strong classification accuracy in distinguishing mucormycosis from other fungal diseases and benign lesions. Earlier studies on deep learning applications in fungal infections have shown that sophisticated convolutional neural networks may identify key traits that standard diagnostic approaches may miss. For instance, machine learning algorithms have shown proficiency in identifying superficial fungal infections using microscopic pictures, highlighting the potential for these technologies to be applied to more intricate illnesses such as mucormycosis [31].

Nonetheless, the shift from conventional biopsy-based diagnostics to deep learning methodologies has some problems. A significant obstacle is the challenge of generalizability across many therapeutic environments. Variations in imaging techniques and the quality of imaging data can impact the performance of deep learning models. A study illustrates that the successful application of deep learning models often depends on access to diverse and robust datasets, which may not be uniformly available in all healthcare facilities [32]. Furthermore, the interpretability of the model’s predictions remains crucial for clinical adoption; physicians want a clear grasp of how models derive their results to create confidence and strengthen diagnostic routes [33].

Another noteworthy problem is the possibility for deep learning algorithms to misclassify cases, particularly in individuals with multilayered clinical presentations or mixed illnesses [34]. When mucormycosis coexists with other opportunistic illnesses, relying entirely on automated methods without supplementary histological confirmation might delay proper diagnosis and potentially worsen patient outcomes.

Therefore, ConvNext Small and related deep learning algorithms should be used as complementary tools within a hybrid diagnostic strategy. They show transformational promise for detecting mucormycosis and reducing reliance on invasive biopsy procedures. This would include merging AI-driven image analysis with conventional diagnostic modalities, including quick molecular testing and focused histopathology assessments, to assure complete, accurate, and efficient diagnoses.

## 5. Conclusions

In this study, we developed and evaluated deep learning models (ResNet50 and ConvNeXt Small) for diagnosing mucormycosis from paranasal sinus CT images. The results indicate that these models can achieve extremely high accuracy, identifying mucormycosis in CT scans with performance that approached perfection on our dataset. This suggests a transformative potential for AI to assist in the early detection of mucormycosis, a critical need given the infection’s rapid progression and high mortality. By substantially speeding up the diagnostic process (from hours/days for biopsy to seconds for AI inference), such models could enable earlier initiation of life-saving treatments and improve patient outcomes.

However, we emphasize that AI tools should act as adjuncts to, not replacements for, established diagnostic methods. In practice, a deep learning model like ours would be best utilized alongside clinical evaluation and conventional tests—for example, to flag high-risk cases for urgent biopsy or to support radiologists in interpreting equivocal imaging findings. We also caution that our model’s current prowess is demonstrated on a limited retrospective dataset. Ongoing and future work will focus on validating this approach in larger multi-center cohorts, including prospective trials to gauge real-world performance. We will also work on improving the model’s generalizability and integrating explainability features to increase clinical trust in the system’s outputs.

Future research avenues include the following: (i) collecting multicenter data to train a more generalized model and testing it against diverse patient populations; (ii) enhancing model interpretability by possibly combining CNNs with symbolic AI or case-based reasoning; (iii) extending the model to handle more categories (e.g., distinguishing mucormycosis from other invasive fungal infections or from bacterial infections), making it a comprehensive tool for sinus disease diagnosis; and (iv) exploring the utility of hybrid models that incorporate clinical parameters (such as patient immune status or lab results) alongside imaging, to improve the diagnostic accuracy further. Additionally, an interesting direction would be to investigate whether deep learning can quantify the extent of disease (for instance, segmenting the areas of fungal invasion) which could help in surgical planning.

In conclusion, our study provides a proof-of-concept that modern AI techniques can reliably detect mucormycosis on imaging. As medicine increasingly embraces digital tools, we foresee that integrating such AI models into routine workflow could help clinicians make faster and more informed decisions in the management of deadly infections like mucormycosis. We must proceed with thorough validation and mindful implementation, but the prospects of improved diagnostic pathways via AI are encouraging, especially in critical care scenarios where every hour counts.

In summary, our work highlights the transformational potential of deep learning algorithms, notably ResNet50 and ConvNext Small, for detecting mucormycosis. However, it emphasizes that new technologies should be adjuncts to established diagnostic approaches rather than substitutes. Ongoing research should focus on strengthening the interpretability of AI systems, broadening training datasets to encompass diverse populations, and verifying these models across various clinical situations to enhance their usefulness in real-world applications.

Future research paths should focus on multicenter studies to evaluate performance across varied populations, increase model explainability, and study the application of these technologies in real-time clinical situations. Thus, as we continue to explore the convergence of technology and medicine, we must embrace these creative tools while retaining a commitment to patient safety and diagnostic accuracy.

## Figures and Tables

**Figure 1 bioengineering-12-00854-f001:**
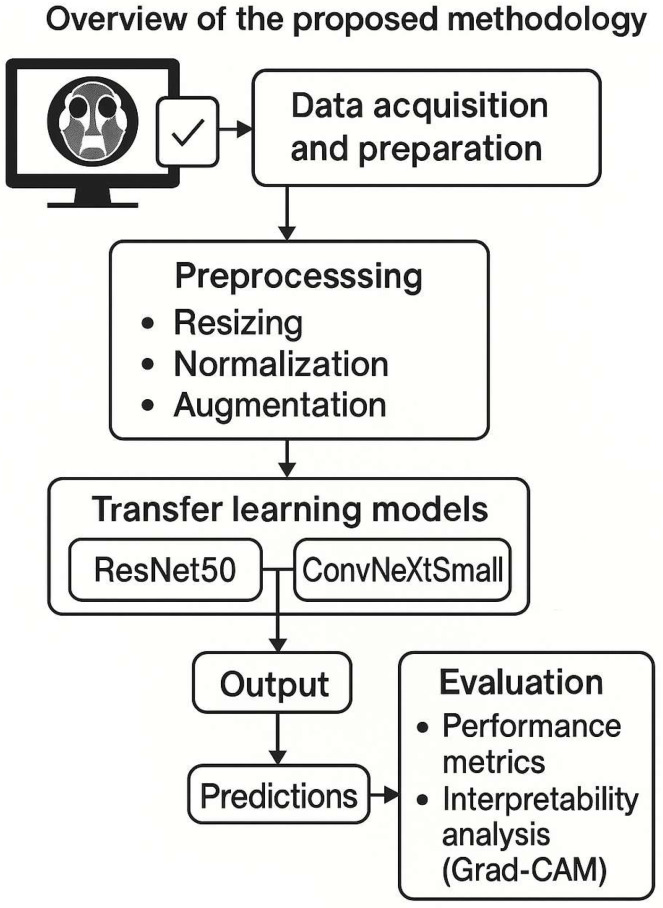
Overview of the proposed methodology. The flowchart depicts the data acquisition and preparation (CT scans collected, labels assigned), preprocessing (resizing, normalization, augmentation), the transfer learning models (ResNet50 and ConvNeXt Small) applied for classification, and the output of predictions. Both models were fine-tuned on our dataset. The diagram also notes the evaluation step, including generation of performance metrics and interpretability analysis (Grad-CAM).

**Figure 2 bioengineering-12-00854-f002:**
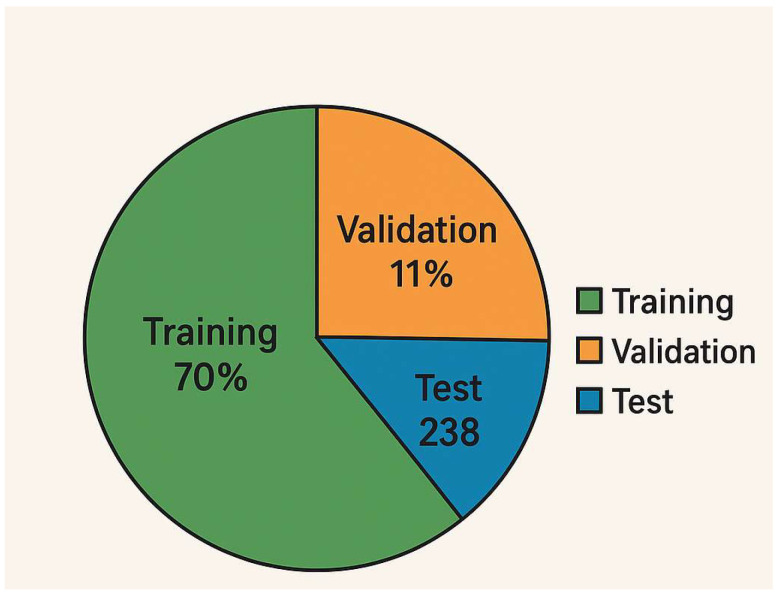
Data split proportions. Visualization of the dataset division, with 70% of images used for model training (with 20% of that for validation) and 30% held for final testing. This resulted in 556 training images (445 after carving out validation), 111 validation images, and 238 test images. The figure confirms that 70/15/30 split (train/val/test) by image count. Each class’s distribution across these sets was stratified to maintain balance (e.g., each fold of cross-validation also maintained class ratios).

**Figure 3 bioengineering-12-00854-f003:**
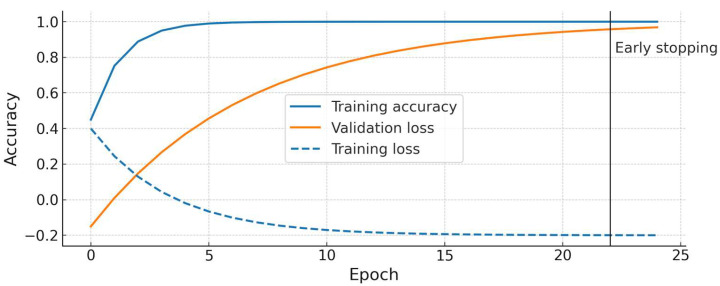
Training and validation accuracy/loss curves for the ConvNeXt Small model. The plot demonstrates that training (blue) and validation (orange) accuracies improve in tandem and reach ~100%, with no significant overfitting gap. The loss curves (dashed lines) decrease and plateau at a low value. Early stopping occurred at epoch 22 in this run once validation loss stopped improving. (ResNet50 showed a similar trend, converging slightly later).

**Figure 4 bioengineering-12-00854-f004:**
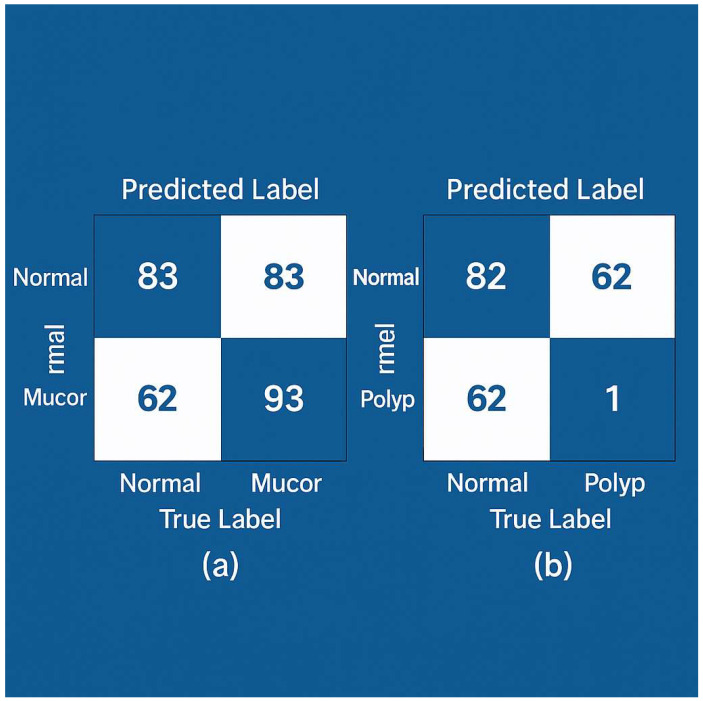
Confusion matrices of model predictions on the test set. (**a**) ConvNeXt Small—all 238 test images are correctly classified (no off-diagonal entries; diagonal values are 83, 62, 93 for Normal, Mucor, and Polyp respectively). (**b**) ResNet50—two misclassifications are observed: one Normal misclassified as Polyp, and one Polyp misclassified as Normal (highlighted in light orange), while all other cases lie on the diagonal. The overall accuracy for ResNet50 is 99.16%. ConvNeXt shows a perfect confusion matrix for this test set.

**Figure 5 bioengineering-12-00854-f005:**
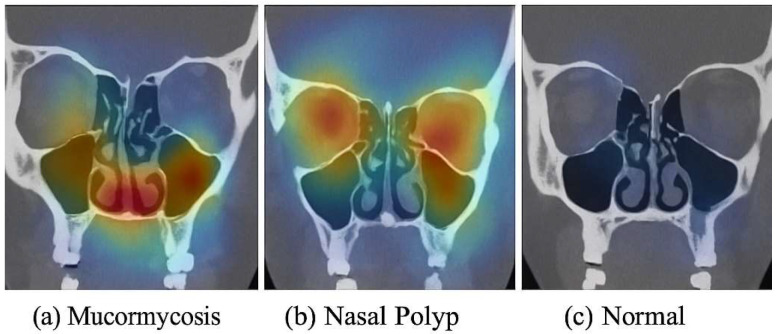
Grad-CAM visualization of model predictions. (**a**) Mucormycosis: CT image (coronal reformat for illustration) from a patient with mucormycosis—the heatmap (yellow/red) highlights the ethmoid sinus region and adjacent orbit where invasive fungal infection was present (the model correctly focuses on the diseased area). (**b**) Nasal polyp: heatmap is concentrated in the nasal cavity where a polypoid mass is seen, matching the lesion’s location. (**c**) Normal: the model’s heatmap shows no specific focal activation (mostly cool colors), consistent with absence of pathology. These examples indicate that the model’s decisions are driven by relevant anatomical abnormalities, providing interpretability to its high performance.

**Table 1 bioengineering-12-00854-t001:** Key hyperparameters.

Parameter	Value
Optimizer	Adam
Learning rate	1 × 10^−4^
Weight decay	1 × 10^−5^
Batch size	16
Max epochs	50
Early stopping	5 epochs without validation loss improvement
Validation split	20% stratified by class

**Table 2 bioengineering-12-00854-t002:** ResNet50 performance metrics on test set.

Class	Precision	Recall	F1-Score	Support (n Images)
Normal	0.99	0.99	0.99	83
Mucormycosis	1.00	1.00	1.00	62
Nasal Polyp	0.99	0.99	0.99	93

All classes show high precision and recall. The slight drop from perfect scores is due to one misclassification in Polyp → Normal and one in Normal → Polyp.

**Table 3 bioengineering-12-00854-t003:** Five-fold cross-validation summary.

Model	Avg. Accuracy (±SD)	Avg. Precision	Avg. Recall	Avg. F1-Score
ConvNeXt	98.9% (±1.1%)	0.99	0.99	0.99
ResNet50	97.8% (±1.5%)	0.98	0.98	0.98

(Averages are macro-averages across classes, per-fold first then averaged. Both models exhibit robust performance across different data splits).

## Data Availability

The raw CT image data for patients used in this study contain protected health information and are not publicly available due to patient privacy regulations. All relevant processed data and results are reported in this article. De-identified data or trained model checkpoints can be shared by the corresponding author upon reasonable request and with permission from the institution, for researchers aiming to validate or replicate the study. The code for the deep learning models and the training pipeline, as well as the trained model weights, will be made openly available on GitHub v1.0 upon publication of this article (repository link to be provided). This will allow independent reproduction of our results and application of the model to other datasets.

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
