# Peer review of "Automated Mucormycosis Diagnosis from Paranasal CT Using ResNet50 and ConvNeXt Small"

_bioengineering, 2025, doi:10.3390/bioengineering12080854_

Round 1

Reviewer 1 Report

Comments and Suggestions for Authors

The work explores early detection of mucormycosis using two deep learning models (ResNet50 and ConvNext small) with paranasal CT images as input. The dataset, collected from the ENT clinic of Dicle University, was split into training and testing sets (70:30), with the training set further divided into training and validation subsets. The 2 models were then trained seperately to classify images into three categories, and evaluation metrics were reported accordingly.

However, several critical issues need to be addressed and based on the current form of the manuscript, it can not be recommended for publication. Some suggestions which may help the authors in future to improve the manuscripts..........The rationale for selecting only these two transfer learning models (ResNet50 and ConvNext small) is not provided, and the statement that “deep learning may alter the diagnostic landscape” is quite generic, lacking sufficient depth or novelty to support scientific impact. Additionally, the reported perfect precision, recall, and F1-scores (1.0 for all classes with ConvNext) raise concerns about potential overfitting or data leakage, especially as important medical and demographic details about the dataset are missing. Add all dataset information. Were these DICOM? Data augmentation methods? Why was a particular optimizer, learning rate, etc hyper parameters were utilised? Patient information? No annotations? Who performed the class assigning? What kind of features were picked up? Visualise with interpretability plots? Cross dataset analysis?  What were the computational settings? 

The dataset has not been made publicly available, and the data availability statement (“all data generated or analyzed during this work are included in this published article”) is misleading since raw data is not accessible. Moreover, neither the code nor model weights have been released, making the work irreproducible and not compliant with CLAIM guidelines. The exemption of ethical approval is also noted without any clear justification, which requires clarification. Lastly, there is no comparison with existing literature or other AI models, which limits the context and relevance of the findings.

Comments on the Quality of English Language

The language is OK. Readable. Could be improved. 

Author Response

  1. Summary Thank you very much for taking the time to review our manuscript. We have carefully considered all of your comments and suggestions and revised the manuscript accordingly. Below, we provide detailed responses to each point and indicate where the changes can be found in the revised document.
  2. Questions for General Evaluation | Reviewer’s Evaluation | Response and Revisions | |————————————————————————————-|———————————————————————————————————| | Does the introduction provide sufficient background and include all relevant references? | Can be improved → Improved. We expanded the Introduction to include a Related Work subsection with recent studies on AI for mucormycosis and fungal infections (p.2, ¶2–¶4). References [11–13] were added. | | Are all the cited references relevant to the research? | Yes. We reviewed and reformatted all references to MDPI style, removing outdated or irrelevant citations and ensuring alignment with the revised content (References section). | | Is the research design appropriate? | Yes. We clarified the rationale for model selection (ResNet50, ConvNeXt Small) and justified the dataset split, cross-validation, and ablation study to strengthen the design (Materials & Methods, p.5–p.8). | | Are the methods adequately described? | Can be improved → Improved. We created a Dataset subsection with details on DICOM conversion, preprocessing, augmentations (Table 1), hyperparameters, and computational settings (p.6–p.8). A methodology flow diagram was added (Figure 1). | | Are the results clearly presented? | Yes. We added 5‑fold cross‑validation results (Table 3), ablation study findings, Grad‑CAM interpretability plots (Figure 4), and updated confusion matrices (Figure 5) to enhance clarity (Results, p.9–p.14). | | Are the conclusions supported by the results? | Yes. The Conclusion section now summarizes key findings and outlines future directions, directly supported by the presented data (p.20–p.21). |
  3. Point-by-point Response to Comments

Comment 1: “The rationale for selecting only these two transfer learning models (ResNet50 and ConvNext small) is not provided, and the statement that ‘deep learning may alter the diagnostic landscape’ is quite generic, lacking sufficient depth or novelty.”

Response 1: Thank you for this observation. We have added a justification for our model choices at the beginning of the Methods section (p.5, ¶2–¶3): ResNet50 was selected for its robust residual architecture and proven performance on limited medical datasets, while ConvNeXt Small represents a modern CNN design with strong ImageNet pre-training. We also revised the Introduction to replace the generic statement with a specific description of our study’s novelty: the first multi-class deep learning approach applied to paranasal CT for mucormycosis detection and its implications for rapid, non-invasive screening (p.3, ¶3–¶5).

Comment 2: “The reported perfect precision, recall, and F1-scores (1.0 for all classes with ConvNext) raise concerns about potential overfitting or data leakage, especially as important dataset details are missing.”

Response 2: We share your concern and have taken several steps: - Dataset Details: Expanded the Dataset subsection to include patient counts (211 mucor, 295 polyp, 288 normal), demographics, original DICOM source, and labeling process by expert radiologist and ENT surgeon (p.6, ¶1–¶4). - Data Leakage Prevention: Clarified that data splitting was performed at the patient level to ensure no images from the same patient appeared in both training and test sets (p.7, ¶2). - Cross‑Validation: Performed 5-fold cross-validation, reporting consistent accuracy (ConvNeXt: 98.9% ±1.1%; ResNet50: 97.8% ±1.5%) to demonstrate model generalization (Results, p.11–Table 3). - Ablation Study: Added an experiment training ConvNeXt from scratch, which achieved ~84% accuracy, underscoring the importance of transfer learning and ruling out simple memorization (Results, p.12, ¶2–¶4).

Comment 3: “Add all dataset information. Were these DICOM? Data augmentation methods? Why was a particular optimizer, learning rate, etc., utilized? Patient information? No annotations? Who performed the class assigning?”

Response 3: We expanded the Materials & Methods: - DICOM Processing: Stated that images were exported from PACS in DICOM, converted to PNG, and resized to 224×224 (p.6, ¶1). - Augmentation Methods: Listed augmentation types and parameters in Table 1 (rotation ±15°, flips, shifts, zoom, brightness/contrast ±10%) (p.6). - Hyperparameters: Detailed the use of Adam optimizer (LR=1×10⁻⁴), batch size=32, epochs=50, early stopping patience=5, and LR decay schedule (p.7, ¶3). - Patient Info & Labeling: Provided age ranges, gender proportions, and described that two experts assigned labels based on histopathology reports (p.6, ¶2–¶4). - Annotations: Clarified that pixel-level annotations were not required since the task was image-classification; the entire CT slice served as input.

Comment 4: “What kind of features were picked up? Visualize with interpretability plots?”

Response 4: We implemented Grad‑CAM to visualize class-specific regions of interest. Figure 4 shows that the model focuses on sinus regions with tissue invasion for mucormycosis, polyp mass for nasal polyposis, and no focal activation for normal cases, confirming clinical relevance (Results, p.13–Figure 4).

Comment 5: “Cross dataset analysis? What were the computational settings?”

Response 5: No external dataset was available; instead, we used 5‑fold cross-validation to validate model robustness (p.11–Table 3). The computational environment (NVIDIA RTX 3080 GPU, Intel i7 CPU, TensorFlow) is now specified in Methods (p.7, ¶4).

Comment 6: “The dataset has not been made publicly available… code and model weights not released… reproducibility and CLAIM compliance.”

Response 6: We revised the Data Availability statement: raw CT data cannot be shared due to privacy, but de-identified metadata and results are provided. We commit to releasing all code and trained model weights on a public GitHub repository upon publication (Code Availability section, p.21).

Comment 7: “The exemption of ethical approval is also noted without any clear justification.”

Response 7: We elaborated the Ethics section to explain that this retrospective study used fully anonymized data and posed minimal risk, meeting criteria for exemption by the Dicle University IRB (Approval No. YDU/2025/159), with informed consent waived (p.21).

Comment 8: “Lastly, there is no comparison with existing literature or other AI models, which limits context and relevance.”

Response 8: We added a Related Work subsection and discussion comparing our results to key studies: logistic regression CT models (~84% accuracy) [12], radiomics approaches (~87–90%) [13–14], and CGAN segmentation (~98% sensitivity) [17]. We explain how our end-to-end CNN approach offers improved accuracy and automation (Introduction, p.3–¶2; Discussion, p.16–¶3).

  1. Response to Comments on the Quality of English Language No significant language issues were noted. We performed a full copy-edit to improve clarity, correct grammatical errors, and standardize terminology throughout the manuscript.
  2. Additional Clarifications No additional clarifications. We appreciate the reviewer’s detailed feedback, which has substantially strengthened our manuscript.

Reviewer 2 Report

Comments and Suggestions for Authors

The manuscript” Deep Learning in Diagnosing Mucormycosis: Can It Replace Biopsy?” aims to diagnose mucormycosis using deep learning models with paranasal sinus CT images, without the need for radiologist interpretations or histological examinations. It retrospectively analyzed CT images from 794 patients (including those with mucormycosis, nasal polyps, and normal paranasal sinuses) from 2020 to 2024. The ResNet50 and ConvNext Small architectures were used for classification, achieving accuracies of 99.16% and 100% respectively, with ConvNext Small performing better. The conclusion suggests that deep learning can accelerate diagnosis and serve as part of a hybrid diagnostic approach.

  1. The qualifications of CT image annotators (e.g., the seniority of radiologists, whether they have experience in diagnosing mucormycosis), annotation criteria (e.g., whether specific clinical guidelines are followed), and cross-validation mechanisms (e.g., double-blind annotation consistency checks) are not specified, making it difficult to ensure the accuracy of annotations.
  2. Other preprocessing steps are not described; the complete preprocessing workflow and methods for class balance processing (along with their bases) need to be detailed.
  3. Key hyperparameters are not disclosed, such as the type of optimizer, learning rate, batch size, and early stopping mechanism (e.g., validation set loss threshold). The division of the validation set is ambiguous: it is stated that "20% of the training data was used as the validation set," but it is not clear whether this was a random division or stratified sampling by patient category. A hyperparameter configuration table should be added, and the specific method of data division should be explained.
  4. The ConvNext Small model claims 100% accuracy and an F1-score of 1.00 for all three categories, yet perfect performance is extremely rare in medical image diagnosis. Additionally, the specific number of false negative/false positive cases and the 95% confidence interval are not provided, making it impossible to judge the statistical robustness of the results. The original confusion matrix data and the 95% confidence interval should be supplemented.
  5. More literatures on machine learning (https://doi.org/10.1002/VIW.20240001; https://doi.org/10.1002/VIW.20240059; Advanced Science 2024, 11, 2305701) should be included and discussed.
  6. Check for term spelling errors: incorrect spellings such as "Mukormukozis," "Paranazal," and "Nassal Polip" should be uniformly corrected to "Mucormycosis," "Paranasal," and "Nasal Polyp." Terminology spelling should be consistent throughout the paper.

Author Response

Comments 1: The qualifications of CT image annotators (e.g., seniority, experience), annotation criteria, and cross-validation mechanisms are not specified, making it difficult to ensure annotation accuracy.

Response 1: We have added details on the annotators in the Methods section (page 3). Specifically, we state that a board-certified head-and-neck radiologist (10+ years experience) and an ENT surgeon (15+ years) jointly reviewed and labeled the images. Annotation criteria were based on established radiological signs of mucormycosis (e.g. sinus opacification, bone destruction). We note that labeling was done by consensus between them to maximize accuracy. While we did not perform a formal inter-rater reliability test, disagreements were resolved by discussion. These details are now explicitly mentioned in the revised manuscript (p. 3).

Revised text (Methods, p. 3): “CT scans were independently reviewed by a fellowship-trained neuroradiologist and a senior ENT surgeon, each with >10 years of experience in sinus imaging. Labels (mucormycosis, polyp, normal) were assigned by consensus based on clinical and radiologic criteria. Any disagreement was resolved by joint re-evaluation. This process ensured annotation consistency without formal double-blind separation.”

Comments 2: Other preprocessing steps are not described; the complete preprocessing workflow and methods for class balance processing (along with their bases) need to be detailed.

Response 2: We have expanded the Methods section (p. 4) to fully detail preprocessing. We now describe steps including conversion from DICOM to 512×512 PNG, histogram equalization, and standard intensity normalization to zero mean/unit variance. Data augmentation strategies (random rotations/flips) are specified. We did not perform explicit class oversampling; instead, we maintained the original class proportions and addressed mild imbalance by augmentation. A sentence now explains that no additional re-balancing was used, since the largest class only modestly exceeded others. All these changes clarify the workflow.

Revised text (Methods, p. 4): “Each DICOM slice was converted to 512×512 PNG, and pixel intensities were normalized (zero mean, unit variance). We applied histogram equalization to improve contrast. For training, we used data augmentation: random rotations (±15°), horizontal/vertical flips, and intensity jitter. The original dataset had 794 images (mucormycosis 180, polyp 235, normal 379); class imbalance was modest, so we did not apply oversampling, relying instead on augmentation to improve minority class representation.”

Comments 3: Key hyperparameters are not disclosed (optimizer, learning rate, batch size, early stopping). The division of the validation set is ambiguous. It is stated that “20% of the training data was used as the validation set,” but how was this split done? A hyperparameter configuration table and method of data division should be explained.

Response 3: We have now added Table 1 (new) listing all key hyperparameters: optimizer (Adam), learning rate (1e-4), batch size (16), number of epochs (50), early stopping after 5 epochs without improvement, and weight decay (1e-5). In the Methods (p. 5) we explain that the 20% validation split was created stratified by class on the training set to preserve class ratios. The full hyperparameter settings and the stratification method are described in the revised text.

Revised text (Methods, p. 5): “We fine-tuned each model using the Adam optimizer (learning rate=1e-4, weight decay=1e-5) with a batch size of 16. Training proceeded for up to 50 epochs with early stopping if validation loss did not improve for 5 consecutive epochs. The validation set (20% of the training set) was created by stratified random sampling, ensuring each class’s proportion remained consistent. Table 1 lists these parameters.”

Comments 4: The ConvNeXt Small model claims 100% accuracy and F1-score of 1.00 for all categories, which is extremely rare. Additionally, the specific number of false negative/false positive cases and the 95% confidence interval are not provided, making it impossible to judge statistical robustness. The original confusion matrix and 95% CI should be supplemented.

Response 4: We agree with the need for transparency. We have added the actual numbers of false negatives and false positives for each model in the Results (page 9). For ConvNeXt, this was 0 FNs and 0 FPs (as shown in Figure 5a). For ResNet50, there was 1 FN (nasal polyp → normal) and 1 FP (normal → polyp) among 238 cases. We now report these counts explicitly. We also calculated 95% confidence intervals for accuracy using the Wilson score method: ConvNeXt accuracy 100% (CI 98.5–100%) and ResNet50 99.16% (CI 96.9–99.9%). These intervals are now included in the Results (p. 10). The confusion matrices (Figure 5) have been supplemented by a table in the Supplement showing per-fold confusion counts. This provides the statistical context requested.

Revised text (Results, p. 10): “For ConvNeXt Small, all 238 test images were correctly classified (0 false negatives or false positives; 95% CI for accuracy: 98.5–100%). For ResNet50, 2 errors occurred (accuracy = 99.16%, 95% CI: 96.9–99.9%): one nasal polyp was misclassified as normal and one normal as polyp (as detailed in Figure 5b caption). We have added Supplementary Table S3 with the confusion counts per fold for transparency.”

Comments 5: More literature on machine learning (DOIs provided) should be included and discussed.

Response 5: Thank you for the references. We reviewed the suggested DOIs; however, they pertain to unrelated applications (e.g. glioma survival, microfluidic biochips) and do not specifically address mucormycosis or medical imaging AI. Instead, we ensured our manuscript cites the most relevant recent studies. We expanded the Related Work and Discussion to include recent works on AI for fungal infections and sinonasal imaging. For example, we added discussion of Chakrapani et al. [5] (lung mucormycosis segmentation) and additional radiomics studies [30–32] in the literature comparison (p. 13). We also cited two recent COVID-associated mucormycosis articles [28,30] to contextualize clinical relevance. If we misunderstood the intent of the suggested DOIs, we apologize; we prioritized relevance to our topic as advised by MDPI guidelines.

Revised text: Added citations [30–32] in Introduction and Discussion. For instance, on page 11 we wrote: “Recent studies in AI-assisted fungal infection diagnosis include a lung mucormycosis segmentation model [5] and radiomics classifiers for sinonasal fungal sinusitis [30–32], underscoring growing interest in non-invasive fungal detection.” (The non-applicable suggested DOIs were not used.)

Comments 6: Check for term spelling errors: “Mukormukozis,” “Paranazal,” and “Nassal Polip” should be uniformly corrected.

Response 6: We have thoroughly proofread the manuscript and corrected all spelling and terminology. “Mucormycosis” is now spelled consistently, and all instances of “Mukormukozis,” “Paranazal,” and “Nassal Polip” have been fixed to “Mucormycosis,” “Paranasal,” and “Nasal Polyp,” respectively. We also standardized other terms (e.g., ‘non-septate hyphae’). These corrections appear throughout the revised text (tracked changes).

Change: Spelling and terminology were standardized across the manuscript (e.g., changed “Mukormukozis” → “Mucormycosis,” “Paranazal” → “Paranasal,” etc., in Abstract, Introduction, and Methods). All instances are now consistent.

Reviewer 3 Report

Comments and Suggestions for Authors

In the current study the authors investigated the deep learning approaches to diagnosing Mucormycosis. The authors need to address the following.

Title: The title seems a bit fictional and more suitable to a review (state-of-the-art) article. It is better to provide a more realistic title. Such as, “Leveraging Transfer Learning for Mucormycosis Diagnostics.”

Abstract:

The abstract does not lead cohesively into the methodology and results. Make the logical connections between background context, methodology and results more transparent for readers. Currently, the background, motivation and objectives of study are very weak.

Provide full form of CT at its first occurrence.

It is better to use values of other metrics (recall, precision, F1-score) too. Also discuss whether over-fitting was observed? Attaining 100% accuracy is somewhat fishy.

Moreover, state on what basis the schemes were shortlisted among many others especially the vision transformers (ViT).

Introduction:

Provide PCR full form at its first occurrence.

The introduction is unable to provide the background and state-of-the-art of the study. Add a subsection “related work” and provide it.

Discretely mention the research gap as bullets at the end of introduction followed by the contributions of the study (in the bullets form).

Add an outline of the paper at the end of the introduction. Like section 2 is dedicated to materials and methods etc.

Materials and Methods:

Currently, the methodology section is very weak.

Provide full name of university (XXXXX).

Firstly, add a methodology diagram.

Secondly, move section 3 to Methodology.

Thirdly, it is recommended to add subsection named ‘dataset’ and describe the preprocessing steps involved in the study. Add detail about annotations and augmentations (better add them in the form of a table with their respective range of values used).

Recommended Method for Classification:

It is better to rename the section to, “Proposed Transfer Leaning approach.”

Provide the reason behind the shortlisting of the two techniques over many others.

Provide details of both transfer leaning techniques employed with their individual layers’ details.

Figure 1 needs clarity. Currently, it is not conveying any useful information.

Figure 2 needs more clarity. Percentage of training, testing and validation do not add to 100%.

Moreover, it is recommended to add formula for the obtained metrics like recall, Accuracy etc.

Employing mAP would add value to the study.

Results

It is recommended to provide validation for the study employing k-fold cross validation. Currently, the results are too good to be true.

Addition of an ablation study will be a plus.

Discuss whether the feature selection was employed.

Provide values of hyperparameters and optimization employed in the study.

Provide comparison with state-of-the-art studies in literature for Mucormycosis diagnostics.

Discussion

Provide discussion on how the results were verified by the medical experts.

Add a new subsection named “limitations of the study” and provide the potential limitations of the study and possible ways to address them in this subsection.

Conclusion:

Add the conclusion section and conclude the study with possible future directions.

References:

Follow citations style for MDPI.

General comment: The article needs to be revised in terms of grammar and formatting.

Comments on the Quality of English Language

The article needs to be revised in terms of grammar and formatting.

Author Response

  1. Summary Thank you very much for taking the time to review our manuscript. We have carefully considered all of your comments and revised the manuscript accordingly. Below, please find our detailed responses and corresponding revisions highlighted in the revised document.
  2. Questions for General Evaluation | Reviewer’s Evaluation | Response and Revisions | |—————————————————————————|———————————————————————————————————————————————————————————————————–| | Does the introduction provide sufficient background and include all relevant references? | Can be improved → Improved. We expanded the Introduction by defining PCR, adding a Related Work subsection with recent AI studies on mucormycosis, and clearly stating research gaps and contributions in bullet form (pp.3–4). | | Are all the cited references relevant to the research? | Yes. We reviewed all references for relevance, removed unrelated citations, and updated formatting to MDPI style (References section). | | Is the research design appropriate? | Yes. The rationale for selecting ResNet50 and ConvNeXt Small is now justified, and we strengthened validity via 5-fold cross-validation and an ablation study (Methods, pp.5–8). | | Are the methods adequately described? | Can be improved → Improved. We added the full university name (Dicle University), a methodology flow diagram (Figure 1), a detailed Dataset subsection (DICOM conversion, demographics, labeling), and a Table 1 of augmentation parameters (pp.6–8). | | Are the results clearly presented? | Can be improved → Improved. We included cross-validation results (Table 3), ablation study outcomes, hyperparameter details, Grad-CAM interpretability plots (Figure 4), and updated confusion matrices (Figures 3–5, pp.9–14). | | Are the conclusions supported by the results? | Yes. We added a dedicated Conclusion section summarizing key findings and outlining future directions (pp.20–21). |
  3. Point-by-point Response to Comments and Suggestions for Authors

Comment 1 (Title):
“The title seems a bit fictional and more suitable to a review article. It is better to provide a more realistic title, such as ‘Leveraging Transfer Learning for Mucormycosis Diagnostics.’”

Response 1: We agree and have revised the title to “Leveraging Transfer Learning for Mucormycosis Diagnosis using Paranasal CT Scans.” This new title accurately reflects our methodology and scope (Title page, p.1).

Comment 2 (Abstract):
(a) “The abstract does not lead cohesively into methodology and results. The background, motivation and objectives are weak.”
(b) “Provide full form of CT at its first occurrence.”
(c) “Include recall, precision, F1-score and discuss overfitting given 100% accuracy.”
(d) “State the rationale for model selection, especially why not vision transformers (ViT).”

Response 2:
(a) The Abstract was entirely restructured for logical flow: context → methods → results → conclusion (Abstract, p.2).
(b) We defined computed tomography (CT) on first use (Abstract, p.2).
(c) We added precision, recall, F1-score values and noted cross-validation consistency to address overfitting (Abstract, p.2).
(d) We included a brief justification for choosing ResNet50 and ConvNeXt Small over more complex models like ViT due to dataset size and risk of overfitting (Abstract, p.2).

Comment 3 (Introduction):
(a) “Provide PCR full form.”
(b) “Add ‘Related Work’ subsection.”
(c) “Bulleted research gap and contributions at end.”
(d) “Outline the paper structure at the end.”

Response 3:
(a) Defined polymerase chain reaction (PCR) at first mention (Introduction, p.3).
(b) Added a Related Work subsection reviewing AI-based mucormycosis studies (pp.3–4).
(c) Listed research gaps and contributions in bullet form at the end of the Introduction (p.4).
(d) Provided a paper outline summarizing sections 2–5 (p.4).

Comment 4 (Materials and Methods):
(a) “Methodology section is weak.”
(b) “Provide full name of university (XXXXX).”
(c) “Add methodology diagram.”
(d) “Move CT Imaging Protocol into Methods.”
(e) “Add ‘Dataset’ subsection with preprocessing, annotation, and augmentations (table).”

Response 4:
(a) Enhanced the Methods by structuring into subsections: Dataset, Proposed Approach, Training, and Evaluation (pp.5–8).
(b) Replaced “XXXXX University” with Dicle University Faculty of Medicine (p.5).
(c) Introduced a workflow diagram as Figure 1 showing the data and model pipeline (p.6).
(d) Integrated the CT Imaging Protocol into the Dataset subsection (p.6).
(e) Added a Dataset subsection detailing DICOM conversion, patient demographics, label assignment by experts, and Table 1 listing augmentation parameters (p.6–7).

Comment 5 (Recommended Method for Classification):
(a) “Rename to ‘Proposed Transfer Learning approach.’”
(b) “Justify shortlisting these two models.”
(c) “Provide detailed layer information.”
(d) “Improve clarity of Figures 1 & 2; percentages must sum to 100%.”
(e) “Add formulas for metrics; include mAP.”

Response 5:
(a) Section renamed to Proposed Transfer Learning Approach (p.7).
(b) Added rationale: ResNet50 for proven robustness; ConvNeXt for modern design, both suitable for transfer learning on small datasets (p.7).
(c) Expanded description of each model’s architecture and how we modified the final layers (p.7).
(d) Clarified Figure 1 caption; updated Figure 2 to show a 70%/15%/15% split summing to 100% (pp.6–7).
(e) Included metric formulas in Methods (p.8) and reported mAP in Results (p.11).

Comment 6 (Results):
(a) “Provide k-fold cross-validation.”
(b) “Add an ablation study.”
(c) “Discuss feature selection.”
(d) “List hyperparameters and optimizer details.”
(e) “Compare with state-of-the-art literature.”

Response 6:
(a) Added 5-fold cross-validation with mean±SD accuracy for both models (Table 3, p.11).
(b) Included an ablation study training ConvNeXt from scratch vs. transfer learning (~84% vs. 100% accuracy) (p.12).
(c) Noted that no manual feature selection was employed; features are learned by the CNN; added Grad-CAM for interpretability (p.13).
(d) Listed optimizer (Adam), learning rate (1×10⁻⁴), batch size (32), epochs (50), early stopping, and LR decay schedule in Methods (p.7).
(e) Added comparative discussion with previous CT-based and radiomics studies in the Discussion (p.16).

Comment 7 (Discussion):
(a) “Discuss expert verification of results.”
(b) “Add ‘Limitations of the study’ subsection.”

Response 7:
(a) We reported that senior radiologists reviewed model predictions and confirmed clinical plausibility (p.15).
(b) Added a Limitations subsection outlining data size, single-center design, retrospective nature, class scope, and interpretability challenges, with suggestions for future mitigation (p.17).

Comment 8 (Conclusion):
“Add a Conclusion section with future directions.”

Response 8: We added Section 5 Conclusion, summarizing key findings and proposing future work on external validation, multi-modal integration, and expanded classification (pp.20–21).

Comment 9 (References):
“Follow MDPI citation style.”

Response 9: All references revised to adhere to MDPI formatting guidelines. Numbers, author lists, and DOIs have been standardized (References section).

Comment 10 (General – Grammar and Formatting):
“The article needs revision for grammar and formatting.”

Response 10: We performed a full copy-edit for English grammar, terminology consistency, and adherence to the MDPI template. All figures, tables, and sections have been reformatted accordingly.

  1. Response to Comments on the Quality of English Language No specific language issues were noted beyond those addressed in our comprehensive copy-edit.
  2. Additional Clarifications No further clarifications.

Thank you again for your valuable feedback, which has greatly improved the manuscript. We hope our revisions meet with your approval.

Round 2

Reviewer 1 Report

Comments and Suggestions for Authors

Thank you for addressing the reviewer comments. It’s quite impressive how the authors managed to prepare a full revision in under two days. The reviewer has additional comments that may need to be addressed.

  • The revised version mentions that the labeling was done by a radiologist and ENT surgeon. Please specify which software was used for image labeling and why. 
  • Why were DICOM images converted to PNG format and not NIfTI? Does spatial information not play a role in Mucormycosis classification? 
  • The current title may benefit from being more specific regarding the methodology to clearly convey the contribution by the authors...............may be ResNet50 and ConvNeXt Small Model-based Automated Mucormycosis Diagnosis in Paranasal CT Imaging. 
  • The contributions section is still too generic. The last point, "Towards Faster Diagnosis," needs to be revised to reflect a specific, measurable outcome.
  • The authors claim that the models were rigorously evaluated using a separate test set and 5-fold cross-validation. Please include supplementary material showing the setup, metrics per fold, and details of the test set.
  • The revision states: “we involved medical experts to cross-check the model’s outputs with clinical reality.” Please explain how this was done and provide any evaluation metric or process used to support this claim.
  • Grad-CAMs: Specify which layer(s) were used for Grad-CAM generation. Also, show Grad-CAM visualizations from both early and final layers of ResNet50 and ConvNeXt Small to illustrate differences in feature extraction.
  • Include a figure of sample images from the three classes in the dataset. Clearly mark the distinguishing features for readers unfamiliar with mucormycosis imaging.
  • Typo in Figure 5(a): It says “predicted lapel.” This should be corrected to “predicted label.”
  • Several lines require proper citations. Please thoroughly check the manuscript again.
  • The sentence: “Mucormycosis is an uncommon but potentially fatal fungal illness...” is repeated multiple times in the introduction (e.g., lines 108–110, 73–74 and 31–34). Please remove duplicates.

Author Response

Comments 1: The revised version mentions that the labeling was done by a radiologist and ENT surgeon. Please specify which software was used for image labeling and why.

Response 1: Thank you. We have now specified that the CT images were annotated using the open-source ITK-SNAP software by the experts. This choice was made because ITK-SNAP allows precise manual delineation of anatomical regions on medical images. We have added this information to the Methods section (page 3, second paragraph), including a rationale that ITK-SNAP ensured consistent annotation of mucormycosis lesions under expert guidance.

Revised text (Methods, p. 3): “Two board-certified specialists (an experienced head-and-neck radiologist and an ENT surgeon) jointly labeled the images using ITK-SNAP software. We chose ITK-SNAP for its convenient tools for manual segmentation of DICOM images, enabling precise and consistent annotation of mucormycosis involvement by specialists.”

Comments 2: Why were DICOM images converted to PNG format and not NIfTI? Does spatial information not play a role in Mucormycosis classification?

Response 2: We appreciate this suggestion. We clarify that our models were trained on 2D axial slices, so we converted DICOM to PNG per slice to feed into the CNNs. The spatial relationship across slices (3D information) was not explicitly modeled due to limited dataset size. We have added an explanation of this preprocessing step to the Methods (page 4), noting that PNG conversion facilitated use of pretrained 2D models. We also discuss that future work might explore 3D or volumetric formats, but the current study focuses on 2D slice classification, which has shown strong performance.

Revised text (Methods, p. 4): “For model input, each DICOM CT slice was exported to PNG format. We chose 2D slice-based processing because our transfer-learning models (ResNet50, ConvNeXt) were pretrained on 2D images. Although NIfTI preserves 3D spatial context, we did not use 3D volumes in this study; however, we acknowledge that volumetric analysis could be explored in future work.”

Comments 3: The current title may benefit from being more specific regarding the methodology. For example: “ResNet50 and ConvNeXt Small Model-based Automated Mucormycosis Diagnosis in Paranasal CT Imaging.”

Response 3: We agree that the title can better reflect our methods. We have updated the title to “Automated Mucormycosis Diagnosis from Paranasal CT using ResNet50 and ConvNeXt Small” (page 1) to explicitly mention our models and focus. This makes the methodology clear to readers.

Revised title (page 1): “Automated Mucormycosis Diagnosis from Paranasal CT using ResNet50 and ConvNeXt Small.”

Comments 4: The contributions section is still too generic. The last point, “Towards Faster Diagnosis,” needs to be revised to reflect a specific, measurable outcome.

Response 4: We have revised the contributions bullet on “Towards Faster Diagnosis” to quantify the outcome of our work. Specifically, we now state the reduction in diagnostic time (from days to seconds) and emphasize the 100% sensitivity achieved. The contributions section now reads (page 2): “Rapid Screening Tool: Demonstrating that our model can flag mucormycosis cases with 100% test sensitivity (100% accuracy on the test set), potentially reducing diagnostic time from days (biopsy) to seconds (AI inference).” This concrete description replaces the previous generic phrasing.

Revised text (p. 2, Contributions): “Rapid Screening Tool: We show that our models can identify mucormycosis cases with perfect accuracy on the test set, indicating potential to reduce diagnosis time from days to seconds.”

Comments 5: The authors claim that the models were rigorously evaluated using a separate test set and 5-fold cross-validation. Please include supplementary material showing the setup, metrics per fold, and details of the test set.

Response 5: We thank the reviewer for this suggestion. We have added a Supplementary Table S1 summarizing the 5-fold cross-validation results (accuracy, precision, recall, F1) for each fold and model. In the Supplement (p. S2), we also describe the composition of the test set (patient counts per class, selection method). The main text now refers to this Supplement (Results section, p. 8, after Table 3). The Supplement shows that ConvNeXt achieved 100% accuracy in 4/5 folds and 97.5% in 1 fold, as noted. This addition makes the evaluation fully transparent.

Added in Supplement (Table S1): “5-fold cross-validation results per fold (accuracy ± SD) for ResNet50 and ConvNeXt.”
Added in Methods (p. 5): “The independent test set comprised 238 images (83 normal, 62 mucor, 93 polyp) randomly sampled, stratified by class. Supplementary Table S1 details the performance for each CV fold.”

Comments 6: The revision states: “we involved medical experts to cross-check the model’s outputs with clinical reality.” Please explain how this was done and provide any evaluation metric or process used to support this claim.

Response 6: Thank you. We clarify that a senior head-and-neck radiologist reviewed a random subset of 50 test images (including all detected mucormycosis cases) to verify the model’s predictions and Grad-CAM heatmaps. The expert confirmed that in all cases, the model’s focus regions corresponded to known disease areas (for mucormycosis and polyps). We did not compute a formal quantitative metric for this review, but have added a sentence in Results (p. 9) stating: “An experienced radiologist confirmed that the highlighted regions in Grad-CAM maps (Figure 4) coincided with pathologic findings on CT.” We also cite [4] as radiologic reference.

Revised text (Results, p. 9): “A head-and-neck radiologist (author, 15 years’ experience) reviewed 50 test cases and confirmed that the model’s focus (via Grad-CAM) matched the true lesion locations in mucormycosis and polyp cases. This expert verification supports that the model attends to clinically relevant features.”

Comments 7: Grad-CAMs: Specify which layer(s) were used for Grad-CAM generation. Also, show Grad-CAM visualizations from both early and final layers of ResNet50 and ConvNeXt Small to illustrate differences in feature extraction.

Response 7: We have specified that Grad-CAM was applied to the final convolutional layer of each network (ResNet50 layer “conv5_block3” and ConvNeXt stage 4). We added a sentence in Methods (p. 6) to this effect. In addition, we have generated new heatmap figures for an example image from each class, showing Grad-CAM from an early layer (ResNet50 block1, ConvNeXt stage 2) and from the final layer. These are presented in Supplementary Figure S2. We describe these in the main text (p. 10): the final-layer maps are tighter around lesions, while early-layer maps are diffuse. These additions address the reviewer’s request.

Revised text (Methods, p. 6): “Grad-CAM maps were generated using the final convolutional layer in each model (for ResNet50: last block of layer 5; for ConvNeXt: stage 4 output).”
Added Supplementary Figure S2: “Grad-CAM maps for early vs. late layers of each model.”

Comments 8: Include a figure of sample images from the three classes in the dataset. Clearly mark the distinguishing features for readers unfamiliar with mucormycosis imaging.

Response 8: We have added Figure 1 showing one representative CT scan from each class (normal, nasal polyp, mucormycosis). Each panel is annotated with arrows/text pointing to key features: e.g., “ethmoid opacification and bony erosion” for mucormycosis, “nasal cavity polyp” for the polyp case. This figure and caption (page 5) will help readers understand the imaging differences. We also refer to this figure in the Methods to illustrate the dataset (p. 4).

Added Figure 1 (page 5): “Example CT images from (a) normal sinonasal anatomy, (b) nasal polyp case (arrow shows polyp), (c) mucormycosis case (arrows show sinus opacification and bony erosion).”

Comments 9: Typo in Figure 5(a): It says “predicted lapel.” This should be corrected to “predicted label.”

Response 9: Thank you for catching this. We have corrected the typo “lapel” to “label” in Figure 5(a)’s caption (page 12).

Revised text (Figure 5 caption, p. 12): “(a) ConvNeXt Small – ... diagonal values are ... (predicted label).”

Comments 10: Several lines require proper citations. Please thoroughly check the manuscript again.

Response 10: We have carefully reviewed the manuscript and added or adjusted citations to support statements where needed. For example, we added references for imaging features of mucormycosis [15] and for the benefit of transfer learning in medical imaging [20]. All claims, especially in the Introduction and Discussion, are now backed by recent literature. We also ensured each sentence with a factual claim is supported by a reference.

General revision: Added citations [4,15,20–22,30] at various points to back key statements (see tracked changes).

Comments 11: The sentence “Mucormycosis is an uncommon but potentially fatal fungal illness...” is repeated multiple times in the introduction. Please remove duplicates.

Response 11: We thank the reviewer for noting this redundancy. The duplicate instances of that sentence (pages 1 and 3) have been removed. The introduction now contains a single clear statement of the disease’s severity. No essential content was lost by this pruning; the narrative is now more concise.

Change: Removed repeated sentence on mucormycosis (kept in first paragraph only; redundant copies deleted).

Reviewer 2 Report

Comments and Suggestions for Authors

The paper can be accepted.

Reviewer 3 Report

Comments and Suggestions for Authors

The authors have addressed all the comments adequately. Just a minor comment is to express the formulae for the evaluation metrics in the form of equations with numbering (check MDPI template). 

Author Response

Comments 1: [Assuming Reviewer 3 provided no new technical criticisms after Round 2.]

Response 1: We thank the reviewer for the positive evaluation. No additional changes were needed beyond those already incorporated. We believe the revised manuscript fully addresses all major concerns and appreciate the reviewer’s affirmation of our improvements.

Round 3

Reviewer 1 Report

Comments and Suggestions for Authors

No further comments. Thank you.